# Oesophageal Atresia: Prevalence in the Valencian Region (Spain) and Associated Anomalies

**DOI:** 10.3390/ijerph20054042

**Published:** 2023-02-24

**Authors:** Adriana Agurto-Ramírez, Laura García-Villodre, Ana Ruiz-Palacio, Berta Arribas-Díaz, Laia Barrachina-Bonet, Lucía Páramo-Rodríguez, Óscar Zurriaga, Clara Cavero-Carbonell

**Affiliations:** 1Service of Preventive Medicine, Valencia General University Hospital Consortium, 46014 Valencia, Spain; 2Rare Diseases Joint Research Unit, Foundation for the Promotion of Health and Biomedical Research in the Valencian Region-Valencia University (FISABIO-UVEG), 46020 Valencia, Spain; 3Public Health Regional Health Administration, Generalitat Valenciana, 46020 Valencia, Spain; 4Department of Preventive Medicine and Public Health, Food Sciences, Toxicology and Legal Medicine, University of Valencia, 46010 Valencia, Spain

**Keywords:** oesophageal atresia, prevalence, risk factors, prenatal diagnosis, epidemiology

## Abstract

The objective was to determine the prevalence of oesophageal atresia (OA) and describe the characteristics of OA cases diagnosed before the first year of life, born between 2007 and 2019, and residents in the Valencian Region (VR), Spain. Live births (LB), stillbirths (SB), and termination of pregnancy for fetal anomaly (TOPFA) diagnosed with OA were selected from the Congenital Anomalies population-based Registry of VR (RPAC-CV). The prevalence of OA per 10,000 births with 95% confidence interval was calculated, and socio-demographic and clinical variables were analyzed. A total of 146 OA cases were identified. The overall prevalence was 2.4/10,000 births, and prevalence by type of pregnancy ending was 2.3 in LB and 0.03 in both SB and TOPFA. A mortality rate of 0.03/1000 LB was observed. A relationship was found between case mortality and birth weight (*p*-value < 0.05). OA was primarily diagnosed at birth (58.2%) and 71.2% of the cases were associated with another congenital anomaly, mainly congenital heart defects. Significant variations in the prevalence of OA in the VR were detected throughout the study period. In conclusion, a lower prevalence in SB and TOPFA was identified compared to EUROCAT data. As several studies have identified, an association between OA cases and birth weight was found.

## 1. Introduction

Oesophageal atresia (OA) is a disorder characterized by an interruption in the continuity of the oesophagus, with or without tracheoesophageal fistula (TEF), which communicates with the trachea [1]. It is the most frequent congenital anomaly (CA) of the oesophagus [2]. CAs are structural or functional abnormalities that are present from birth, although they can manifest at later periods, and constitute a diverse group of conditions of prenatal origin that may be due to single gene defects, chromosomal abnormalities, multifactorial inheritance, environmental teratogens, or lack of micronutrients [3].

Most CAs are considered rare diseases due to their low prevalence (in Europe, less than five cases per ten thousand inhabitants). Rare diseases, including those of genetic origin, are chronically debilitating, disabling, and even life-threatening conditions [4]. The global incidence of OA varies from 1/2500 to 1/4500 live births (LB) [2]. The prevalence of OA for the period 2007–2019 in the European network of population registries for the epidemiological surveillance of CA (EUROCAT) is 2.63/10,000 births, remaining stable during the last decades but with a slight variation between European regions [5].

OA can be present in association with other CAs, generally those included in the VACTERL association, such as vertebral defects, anal atresia, cardiac malformations, TEF, renal anomalies, and limb malformations [6]. The etiology of this disease remains unknown, although it has been linked to genetic and environmental factors. The most associated genetic factors are trisomies, such as Down, Edwards or Patau syndrome, as well as alterations of a single gene, such as CHARGE and Feingold syndromes or Fanconi anemia, among others. Among the environmental factors, maternal exposure to drugs such as alcohol and tobacco, the use of in vitro fertilization techniques and gestational diabetes mellitus stand out [7].

OA is differentiated into types depending on its location and the presence or absence of TEF. The tenth revision of the international classification of diseases with the extension of the British Pediatric Association (ICD10-BPA) used by EUROCAT, classifies OA as OA without mention of a fistula or without other specification (code Q39.0) and OA with TEF (code Q39.1). According to Vogt’s classification [8], in 86% of the cases of type III OA or with distal TEF are detected, in 7%, type I or without associated TEF, in 4%, type V or with TEF without atresia and, with less frequency, type II OA or with proximal TEF, and type IV or with proximal and distal TEF (<1%) [8]. A comparison of both classifications is presented in Figure 1.

The diagnosis is usually done in the first 24 h of life and may be suspected in the presence of hypersalivation or the inability to swallow saliva. During the prenatal stage, the presence of polyhydramnios or the absence of stomach bubbles, normally observed between 16 and 20 weeks of gestation (GW), can be considered predictive factors. Another warning sign is the dilation of the atretic blind fundus detected during swallowing in the third-trimester ultrasound [9]. Diagnosis confirmation is obtained by a chest and abdominal X-ray demonstrating the abnormality [10].

Case mortality is directly related to low birth weight and major congenital heart defects [1]. Factors such as prematurity can have a negative influence, increasing the mortality of cases; however, the presence of TEF and the existence of other associated anomalies have not been shown to increase mortality [11].

The CA population-based registry of VR (RPAC-CV), which is part of EUROCAT [12], collects information on those diagnosed with OA before the first year of life, and who are residents in the VR. Based on these data, a study was performed to determine the prevalence of OA in the VR and describe the characteristics and distribution of cases with OA born between 2007 and 2019 in the VR.

## 2. Materials and Methods

A cross-sectional study was performed on cases diagnosed with OA before the first year of life, with or without TEF, born during the period 2007 and 2019 in the VR. The VR is one of seventeen regions of Spain, with a population of approximately 5 million and an annual number of births around 45,000.

The RPAC-CV was used as a source of information, from which the cases with a confirmed diagnosis of OA were obtained, coded with the codes Q39.0 and Q39.1 of the ICD10-BPA. The inclusion criteria used were those marked by EUROCAT that consider as cases all those residing in the VR who present at least one major CA [12]. The study subjects were LB, stillbirths (SB), and termination of pregnancy for fetal anomaly (TOPFA), diagnosed prenatally or during the first year of life.

The variables included in the analysis were those related to the case, to the CA, and to the pregnant woman.

-Case variables: Year of pregnancy ending, type of pregnancy ending (LB, SB, or TOPFA), sex (male, female, or unknown), gestational age, weight, and the number of babies at the time of pregnancy ending, date of death in LB cases, and surgery performed during the first year of life (yes/no/unknown).-CA variables: Type of CA, other CA and associated syndromes, time of diagnosis, and diagnostic gestational age of the first CA in cases with prenatal detection.-Pregnant woman variables: Assisted conception (yes/no/unknown), previous and during pregnancy medical history, history of previous spontaneous abortions or previous TOPFA, medication during the first trimester of pregnancy, prenatal tests performed, country of birth, and province of residence (Castellón, Valencia and Alicante).

Regarding the statistical analysis, firstly, the prevalence per 10,000 births and its 95% confidence intervals (95%CI) were calculated for the whole period and for each year. In addition, the prevalence by the type of pregnancy ending was calculated. The distribution of cases by sex and weight at birth was obtained. Birth weight in LB was divided according to the classification recommended by the World Health Organization (WHO) [13]: very low birth weight (VLBW) if ≤1500 g, ≤2500 g low birth weight (LBW), >2500 g–3999 g normal weight, and ≥4000 g macrosomic [13], and the mean weight of LB cases at birth was obtained.

The frequency of OA was described according to the number of babies at the pregnancy ending, as well as by gestational age, classifying the cases as less than 28 GW, 28–32 GW, 33–36 GW, and 37 GW or more [14]. The mean gestational age at pregnancy ending was obtained for all cases, including SB and TOPFA.

In LB cases who died during the first year of life, the time elapsed from birth to death was calculated, obtaining the median of the days elapsed. The overall crude mortality rate out of 1000 births and groups according to birth weight were obtained. In addition, the frequency of cases that required some surgical procedure during the first year of life was calculated. A Fisher exact test was carried out to study the relationship between birth weight categories and death of the cases (yes/no).

Moreover, the frequency of cases according to the type of OA was determined. Once the RPAC-CV cases coded according to the ICD10-BPA were obtained, they were adapted to the Vogt’s classification [8] using the literal of the diagnosis that includes the location of the TEF (type V was not taken into account in this study because it does not include OA diagnosis). In addition, the CA and syndromes most frequently associated with OA were identified and grouped by subgroups according to EUROCAT [12], and their frequency was analyzed. The frequency of cases was calculated according to the time of diagnosis, as well as the mean gestational age of the first CA in cases with prenatal detection.

The number of cases conceived by assisted conception and the frequency of pregnant women with a history of spontaneous abortions and previous TOPFA were studied, as well as the frequency of maternal diseases before and during pregnancy. The drugs used during the first trimester of pregnancy were classified according to the groups of the anatomical-therapeutic-chemical (ATC) classification [15], and the country of birth of each pregnant woman was determined.

Finally, the distribution of cases and the prevalence according to the mother’s residence by provinces of the VR were analyzed to describe their geographical distribution.

Statistical analysis was performed using the IBM SPSS Statistics 22 software by applying the chi-square statistical test for qualitative variables and Student’s T test for quantitative variables, to detect statistically significant differences.

## 3. Results

A total of 146 OA cases were identified during the period from 2007 to 2019 in the RPAC-CV. The overall period prevalence was 2.4/10,000 births (95%CI: 2.0–2.8), being 2016 the year with the highest prevalence (3.8/10,000 births), and 2014 the one with the lowest (1.6/10,000 births). Figure 2 shows the evolution of the annual prevalence for each year of the study period.

Regarding the distribution by type of pregnancy ending, 97.3% (142 cases) of the OA cases were LB, 1.4% (2 cases) were SB, and 1.4% (2 cases) corresponded to TOPFA. The prevalence by type of pregnancy ending was 2.3/10,000 births (95%CI: 2.0–2.7) for LB and 0.03/10,000 births (95%CI: 0.0–0.1) for both SB and TOPFA. Table 1 shows the annual prevalence of OA cases, with or without TEF, according to the type of pregnancy ending.

The distribution by sex of the OA cases was 57.5% male and 41.1% female. In 1.4% of the cases, corresponding to TOPFA, this information was unknown.

Regarding the weight of the LB cases, a mean birth weight of 2426 ± 0.963 g was obtained. In addition, 10.3% had VLBW, 41.8% LBW, 43.8% normal weight, and 0.7% were macrosomic. In 0.7% of cases, this information was unknown.

It was observed that 6.8% of the cases corresponded to twin pregnancies and 2.1% to triple gestations. The rest of the cases were single pregnancies (91.1%). The mean gestational age at the time of pregnancy ending was 36.7 ± 6.3 GW, and it was identified that 47.3% of cases were between 33–36 GW, 38.4% 37 GW or more, 10.3% between 28–32 GW and, finally, 3.4% less than 28 GW. The GW at pregnancy ending was unknown in 0.7% of the cases.

Considering only LB cases, 13.7% died before one year of age. The median number of days elapsed between birth and death was 6 days. A crude mortality rate during the first year of life of 0.03 per 1000 births was observed for the period 2007–2019. In relation to the weight at birth, of the 20 cases dead, 85.0% had VLBW or LBW (Table 2). Mortality in LBW cases was higher than in normal weight cases, with a crude mortality rate of 0.01 per 1000 births for VLBW, 0.02 per 1000 births for LBW, and 0.005 per 1000 births for normal weight. The mortality rate during the period is shown in Figure 3. A Fisher’s exact test was performed to study the relationship between the weight categories at birth and the death of cases, obtaining a statistically significant association (*p* < 0.05).

In 88.7% of LB cases, at least one surgical procedure was performed during the first year of life, while surgery was not required in 1.4%, and in 0.7% surgery was not needed because it was considered too severe for the procedure. In 9.2%, this information was unknown.

Concerning the type of OA, according to Vogt’s classification [8] or location of the TEF, a higher frequency of OA with distal TEF or type III was detected, followed by OA without TEF or type I (Table 3).

Of the 146 cases, 71.2% had another CA associated. A total of 334 associated malformations were identified since more than one different associated anomaly was identified in some of the cases studied. Most of these malformations corresponded to congenital heart defects (Table 4). The relationship between congenital heart disease (yes/no) and case mortality (yes/no) was studied using the chi-square test, where no statistically significant relationship was obtained (*p* > 0.05).

In addition, OA was detected to be associated with syndromes or associations of malformations in 15.8% of the total cases. Among these, the most frequent was the VACTERL association (in 43.5% of cases), followed by Edwards syndrome (26.1%). In third place, the Polymalformative syndrome (13.0%) was found, and finally, the Patau, Crouzon, Cri-du-chat syndromes, and the CHARGE association, with 4.3% each.

According to the time of diagnosis of the first CA in each case (it can be the OA or another associated CA), in 58.2% of cases it was detected at birth, and in 37.7% it was diagnosed prenatally. In 2.1%, it was seen during the first week of life, and in 2.1% of the cases this information was unknown.

In those diagnosed prenatally, ultrasound was the predominant diagnostic technique during the prenatal stage. In 43.6% of cases, the first malformation was detected in the third trimester of pregnancy, 30.9% during the second trimester, and 3.6% during the first trimester (Table 5). In 21.8% of cases, this information was unknown. The mean gestational age at prenatal diagnosis was 27.2 ± 6.5 GW.

The mean age of the pregnant women at the time of pregnancy ending was 32 years (with a range of 18 to 47 years). A total of 13.0% of the pregnancies were conceived by assisted conception. The relationship between assisted conception and the type of OA was studied using the chi-square test, where no statistically significant differences were identified (*p* > 0.05). On the other hand, 21.2% of the pregnant women had a history of spontaneous abortions and 11.0% of previous TOPFA.

Endocrine diseases (11.6%), such as hypothyroidism, obesity, and hyperlipidemia, were the most frequently observed medical diseases before pregnancy in the pregnant women, followed by a personal history of CA (5.5%), such as kidney abnormalities disease, congenital heart disease, and pleural abnormalities. Likewise, gynecological pathologies (4.8%), infections (4.1%), hereditary genetic diseases (3.4%), respiratory (2.0%), psychiatric, vascular, digestive, and allergies were found, with a frequency of 1.3% each.

In addition, 37.7% of the pregnant women presented some pathologies during pregnancy. Specifically, a total of 67 diseases were diagnosed, in some cases the pregnant women had more than one disease. Polyhydramnios (20.0%), gestational diabetes mellitus (18.5%), hypothyroidism (10.8%), urinary tract infections (10.8%), and gestational hypertension (7.7%) were more frequently observed. A total of 26.7% of the pregnant women did not have gestational diseases, and there was no information available in 35.6% (Table 6).

A total of 37.0% of the pregnant women took drugs during the first trimester of pregnancy, while in 45.9%, this information was not available. The most used drugs were antibiotics, mainly clindamycin in suppositories and ampicillin, followed by antithyroid drugs, vitamin and pregnancy supplements, corticosteroids, and antihypertensives (Table 7).

Regarding the country of birth of the pregnant women, 56.8% were Spanish born and 17.8% were foreigners. Among the most frequent foreign countries of birth were Moroccan, Romanian, and Bolivian origin. The country of origin was unknown in 25.4% of the pregnant women.

Regarding the geographical distribution according to the maternal residence, it was observed that 49.3% of the pregnant women resided in the province of Valencia, 43.8% in the province of Alicante, and 6.8% in the province of Castellón. The prevalence by provinces for the study period was 2.9/10,000 births (95%CI: 2.2–3.7) in Alicante, 2.3/10,000 births (95%CI: 1.8–2.8) in Valencia, and 1.4/10,000 births (95%CI: 0.5–2.2) in Castellón.

## 4. Discussion

The overall prevalence of OA cases obtained in the VR for the period 2007–2019 was more similar than EUROCAT [5]: 2.3/10,000 births for the same period. It was also more similar than other population-based registries of CA which belong to EUROCAT [5], such as the Basque Country (Spain), whose prevalence was 2.5/10,000 births. In the case of Norway, a European country whose population is quite comparable to that of VR and is also part of EUROCAT [5], we could find a prevalence only slightly higher (2.8/10,000 births) than that in VR [5].

Furthermore, studies such as the one by Nassar et al. [16], whose cases, classified using the ICD9-BPA or ICD10-BPA which belonged to members of birth defects surveillance programs in North America, South America, Europe, and Australia, found a global prevalence of OA similar to that obtained in the RPAC-CV: 2.4/10,000 births during the period 1998–2007 [16].

In VR, significant variations were detected in the annual prevalence of OA cases throughout the study period, with the lowest prevalence being in 2014 and the highest in 2016.

The prevalence by type of pregnancy ending of OA cases identified in EUROCAT [5] during the period 2007–2019 was higher than that obtained in the VR both in SB (0.06/10,000 births) and in TOPFA (0.13/10,000 births).

Concerning the sex of OA cases in the VR, a slight male predominance was detected, in agreement with what was described in the work of Vara Callau et al. [8], in which a ratio of 1.5:1 was found. However, the frequency of twin and triple pregnancies was lower in the VR than that found by these authors [8]. 

Regarding the time of diagnosis in cases of OA of the VR, 37.7% were detected prenatally, this value being higher than that found by Sfeir et al. [11], which described, for the period 1998–2007, a prenatal diagnosis in 30% of cases [11]. It is important to remark that Sfeir describes only prenatal OA diagnosis, and in VR, any first CA diagnosis is included, which is not necessarily OA. Advances in the technique applied to prenatal tests may be the reason for this increase. On the other hand, coincidences have been found in the time elapsed until the moment of diagnosis in the cases detected postnatally, both being within the first 24 h of life [17].

The mean gestational age at the time of pregnancy ending in OA cases in the VR was slightly lower than that described by Vara Callau [8], 36.7 ± 6.3 GW vs. 37.1 ± 2.6 GW, respectively [8]. A total of 61.0% of OA cases of the RPAC-CV ended the pregnancy with less than 37 GW, a much higher frequency than that found in the general population, where a prematurity rate of 8.3% had been described [18]. A total of 52.1% of the cases were low birth weight (including VLBW and LBW), a higher value than that described by Galarreta et al. [19], where it was found that 49.6% of the cases were VLBW and LBW at birth [19]. When comparing the frequency of cases with VLBW, 10.3% found in our sample contrasts with 8.6% described in the aforementioned study [19].

Moreover, in OA cases in the VR, a higher percentage of male sex, low weight, and preterm gestational age (≤36SG) at the time of pregnancy ending were identified in comparison to all the cases with AC from the RPAC-CV during the same period of study [20].

According to others studies [1,21], the mortality of OA cases is directly related to birth weight and associated heart defects. In the VR, statistically significant differences in birth weight and mortality in LB cases were found. Congenital heart defects were the most frequent ones associated with OA in VR. However, no statistically significant differences were found between congenital heart disease and OA cases mortality [1].

The association of OA with other CAs has been repeatedly described in different studies [1,22], suggesting the need to look for associated malformations before diagnosing OA. A higher frequency of CA associated with OA was found in VR (71.2%) compared to the 50% described by De Jong et al. [7]. Among the associated CA, the author [7] describes a frequency of 10% of cases related to some component of the VACTERL association, a higher frequency than that found in VR (6.2%), although also prevailing over other associated syndromes.

In the OA cases of VR, 15.8% were associated with syndromes or associations of malformations and 6.9% with chromosomal abnormalities, equivalent to that described in the literature [22] and lower than that observed by Galarreta et al. [19], which describes 10.2% of cases associated with chromosomal abnormalities. The main chromosomal abnormality related to the RPAC-CV cases was Edwards syndrome (4.1%), with a lower frequency than that described by Felix et al. [23] but prevailing over the rest of the chromosomal abnormalities.

In VR, the frequency of cases with type III OA was lower than in similar studies [9]; however, the frequency of cases with OA type I was higher than that found in these studies [9]. This may be due to the fact that we have a high percentage of OA cases with TEF without specifying the location, and, according to the literature [9], it would be expected that they were mainly type III.

In addition, in pregnant women with OA cases from the RPAC-CV, 21.2% of the history of previous spontaneous abortions and 11.0% of the history of previous TOPFA were found, coinciding with the 20% of the history of spontaneous abortions that are described in the literature [24] and with 11.7% of the history of TOPFA in Spanish public hospitals in 2015 [25]. 

Gestational diabetes mellitus has been associated with the appearance of CA, macrosomia, neonatal complications, and a high percentage of perinatal mortality [17]. In VR, it was found that 8.2% of pregnant women with OA cases developed gestational diabetes during pregnancy, a higher incidence than that described in the literature [17], where an incidence between 1% and 5% of pregnancies was estimated [17]. In addition, a 5.5% personal history of CA was observed in pregnant women, a percentage that coincides with that described by Spitz [26], who also describes a similar proportion of a history of CA in first-degree relatives with one or more components of the VACTERL association [26].

The limitations of the study may be due to the small number of cases intrinsically associated with OA as it is a rare disease, which could only be expanded by studying a more extended period or expanding the study territory. Another limitation is the lack of information found in some of the clinical variables under study, which will foreseeably improve over time since the recent implementation of the electronic medical record in the Spanish health system, which seems to be increasing the quality of health data and its collection [16].

## 5. Conclusions

In conclusion, the global prevalence of OA cases obtained in the RPAC-CV was similar to EUROCAT (2.3/10,000 births) for the same period. However, EUROCAT identified a higher prevalence in SB (0.06/10,000 births) and TOPFA (0.13/10,000) than those obtained in VR. OA is a CA whose mortality is influenced by factors such as birth weight. In many cases, the OA is associated with other CAs, mainly congenital heart defects. The appearance of TEF is quite frequent, being type III OA the one that prevails. Although the prenatal diagnosis of OA has increased over time, detection at birth continues to be more frequent.

## Figures and Tables

**Figure 1 ijerph-20-04042-f001:**
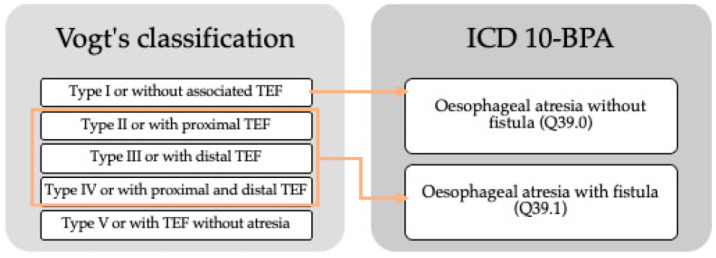
Comparison of Vogt’s classification and ICD10-BPA of oesophageal atresia. TEF: tracheoesophageal fistula.

**Figure 2 ijerph-20-04042-f002:**
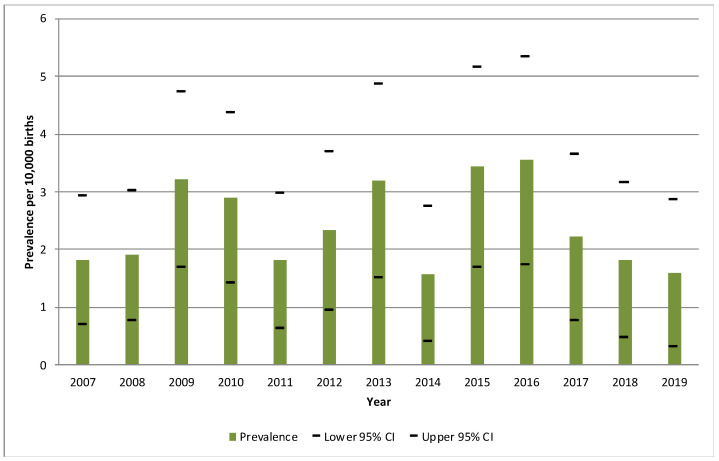
Annual prevalence of cases with oesophageal atresia per 10,000 births and their 95% confidence intervals during the period 2007–2019, in the Valencian Region. CI: confidence interval.

**Figure 3 ijerph-20-04042-f003:**
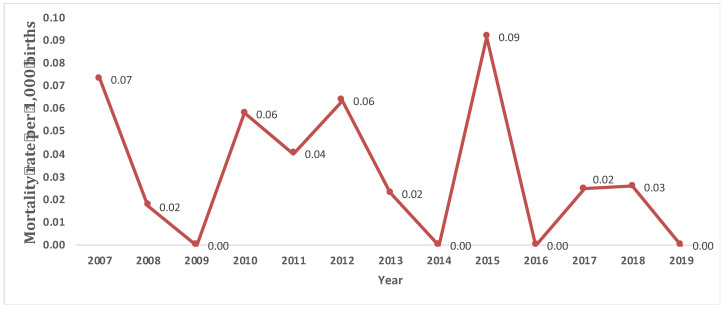
Evolution of the annual mortality rate (per 1000 births) in live-birth cases with oesophageal atresia during the period 2007–2019 in the Valencian Region.

**Table 1 ijerph-20-04042-t001:** Annual prevalence of cases with oesophageal atresia per 10,000 births and their 95% confidence intervals according to the type of pregnancy ending during the period 2007–2019 in the Valencian Region.

	Prevalence (95%CI)		
Year	Live Births	TOPFA	Stillbirths
2007	1.8 (0.7–2.9)	0.0	0.0
2008	1.9 (0.8–3.0)	0.0	0.0
2009	3.2 (1.7–4.7)	0.0	0.0
2010	2.9 (1.4–4.4)	0.0	0.0
2011	1.8 (0.6–3.0)	0.0	0.0
2012	2.3 (1.0–3.7)	0.0	0.0
2013	2.7 (1.2–4.3)	0.2 (−0.2–0.7)	0.2 (−0.2–0.7)
2014	1.6 (0.4–2.7)	0.0	0.0
2015	3.0 (1.4–4.6)	0.2 (−0.2–0.7)	0.2 (−0.2–0.7)
2016	3.8 (1.9–5.6)	0.0	0.0
2017	2.2 (0.8–3.7)	0.0	0.0
2018	1.8 (0.5–3.2)	0.0	0.0
2019	1.6 (0.3–2.9)	0.0	0.0

TOPFA: termination of pregnancy for fetal anomaly, 95% CI: 95% confidence interval.

**Table 2 ijerph-20-04042-t002:** Distribution of live-birth cases with oesophageal atresia, according to weight at the pregnancy ending and number of deaths at one year of life during the period 2007–2019 in the Valencian Region.

Birth Weight (g)	Number of Cases	Number of Deaths (Percentage by weight Category)
Very low weight (≤1500)	15	6 (40.0)
Low weight (≤2500)	61	11 (18.0)
Normal weight (>2500–3999)	64	3 (4.7)
Macrosomal (≥4000)	1	0 (0.0)
Unknown	1	0 (0.0)
Total	142	20 (28.2)

**Table 3 ijerph-20-04042-t003:** Distribution of cases with oesophageal atresia according to Vogt’s classification or location of the fistula during the period 2007–2019 in the Valencian Region.

Type of Oesophageal Atresia	Number of Cases	Percentage (%)
OA type III (distal TEF)	70	47.9
OA type I (without TEF)	31	21.2
OA type II (proximal TEF)	3	2.1
OA with TEF NE	42	28.8
Total	146	100

OA: Oesophageal atresia. TEF: Tracheoesophageal fistula. NE: Not specified.

**Table 4 ijerph-20-04042-t004:** Distribution by groups of malformations associated with oesophageal atresia during the period 2007–2019 in the Valencian Region.

Group of Malformations Associated	Number of Cases	Percentage (%)
Congenital heart defects	144	43.1
Digestive system (excluding OA)	36	10.8
Urinary system	25	7.5
Respiratory system	23	6.9
Chromosomal	23	6.9
Limbs	17	5.1
Other abnormalities or syndromes	15	4.5
Peripheral vascular system	14	4.2
Genital organs	12	3.6
Nervous system	9	2.7
Eyes	6	1.8
Ears, face, and neck	5	1.5
Skeletal	3	0.9
Cleft lip and cleft palate	2	0.6
Total	334	100

OA: Oesophageal atresia.

**Table 5 ijerph-20-04042-t005:** Distribution of cases with oesophageal atresia according to the time of diagnosis during the period 2007–2019 in the Valencian Region.

Time of Diagnosis	Number of Cases	Percentage (%)
At birth	85	58.2
Prenatal	55	37.7
First trimester	2(3.6%)	
Second trimester	17 (30.9%)	
Third trimester	24 (43.6%)	
Unknown	12 (21.8%)	
During the first week of life	3	2.1
Unknown	3	2.1
Total	146	100

**Table 6 ijerph-20-04042-t006:** Distribution of gestational diseases of the pregnant women of cases with oesophageal atresia during the period 2007–2019 in the Valencian Region.

Diseases	Number	Percentage (%)
Polyhydramnios	13	19.4
Gestational diabetes mellitus	12	17.9
Hypothyroidism	7	10.4
Urinary tract infection	7	10.4
Hypertension	5	7.5
Preeclampsia	3	4.5
Placental disorders	3	4.5
Preterm labor without delivery	2	3.0
Hepatic and biliary tract disorders	2	3.0
Threatened abortion	2	3.0
Parasitic infections	1	1.5
Anemia	1	1.5
Beta thalassemia	1	1.5
Insufficient fetal growth	1	1.5
Incompetence of the cervix	1	1.5
Oligohydramnios	1	1.5
Premature rupture of membranes	1	1.5
Bipolar disorder	1	1.5
Varicose veins in lower extremities	1	1.5
Not specified	2	3.0
Total	67	100

**Table 7 ijerph-20-04042-t007:** Distribution of drugs used during the first trimester of pregnancy in the pregnant women of cases with oesophageal atresia according to the ATC classification during the period 2007–2019 in the Valencian Region.

Drugs Used in 1st Trimester	Number	Percentage (%)
Antibiotics	24	26.4
Antithyroid	16	17.6
Corticosteroids	13	14.3
Vitamins and supplements	13	14.3
Antihypertensives	9	9.9
Anticoagulants	4	4.4
Antidepressants	3	3.3
Anti-D immunoglobulin	2	2.2
Progestogens	2	2.2
Antihistamines	1	1.1
Biological drugs	1	1.1
Other gynecological preparations	1	1.1
Antacids	1	1.1
Beta-2 Adrenergic Receptor Agonists	1	1.1
Total	91	100

## Data Availability

The data that support the findings of this study are available from Information System on Rare Diseases in Valencian Region (Spain).

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
