# Peer review of "Oesophageal Atresia: Prevalence in the Valencian Region (Spain) and Associated Anomalies"

_ijerph, 2023, doi:10.3390/ijerph20054042_

Round 1
Reviewer 1 Report
I enjoy reading this paper that, although not original, it is well written, statistics is good, and precise information are delivered regarding a selected population.
Some minor corrections are required:
line 237 ("...with syndromes in 15.8% of the total cases. Among these, the most frequent was the VACTERL association....") . VACTERL is not a syndrome, but an association of malformations, therefore the above sentence sholud be rephrased
line 247-251: do the Authors refer to the prenatal diagnosis of OA (and in this case 37,7% of prenatal diagnosis sounds too high, expecially during the first and second trimester) or to associated malformations (and in this case the whole sentence needs to be rephrased)?
324 Regarding the time of diagnosis in cases of OA of the VR, 37.7% were detected prenatally. Please, see the previous comment.
303: please, add the following reference, as a similar and recent work has been published (Prevalence, characteristics, and survival of children with esophageal atresia: a 32-year population-based study including 1,417.724 consecutive newborns Birth Defects Res A Clin Mol Teratol. 2016 Jul;106(7):542-8. doi: 10.1002/bdra.23493. Epub 2016 Mar 2.)
Some minor spell ckecks are required (i.e. line 269 "some pathology", 270 Specifically, a total of 67 diseases were diagnosed, finding, ....; 335 had low birth weight, 337 had VLBW and LBW, 338 the 10.3% found in our sample contrasts with the 8.6% described in...., 368 a 21.2%, 369 an 11.0%, the 20%)
Author Response
First of all, thanks for the comments about the paper. Regarding the review, we responded to your comments.
Point 1: line 237 ("...with syndromes in 15.8% of the total cases. Among these, the most frequent was the VACTERL association....") . VACTERL is not a syndrome, but an association of malformations, therefore the above sentence sholud be rephrased
Response 1: We have corrected the text as suggested. We rephrased the sentence as “with syndromes or associations of malformations” (we did the change in line 237 and in line 355).
Point 2: line 247-251: do the Authors refer to the prenatal diagnosis of OA (and in this case 37,7% of prenatal diagnosis sounds too high, expecially during the first and second trimester) or to associated malformations (and in this case the whole sentence needs to be rephrased)?
Response 2: We apologize for the current ambiguous text, we refer to the diagnosis of the first congenital anomaly, and it can be OA or another CA associated
Point 3: 324 Regarding the time of diagnosis in cases of OA of the VR, 37.7% were detected prenatally. Please, see the previous comment.
Response 3: As in a previous response, we rephrased the paragraph of discussion regarding the time of diagnosis to clarify these.
Point 4: 303: please, add the following reference, as a similar and recent work has been published (Prevalence, characteristics, and survival of children with esophageal atresia: a 32-year population-based study including 1,417,724 consecutive newborns Birth Defects Res A Clin Mol Teratol. 2016 Jul;106(7):542-8. doi: 10.1002/bdra.23493. Epub 2016 Mar 2.)
Response 4: Thank you for your suggestion, we added the reference and we compared our results with this study in our discussion. Consequently, we also renumber the references.
Point 5: Some minor spell ckecks are required (i.e. line 269 "some pathology", 270 Specifically, a total of 67 diseases were diagnosed, finding, ....; 335 had low birth weight, 337 had VLBW and LBW, 338 the 10.3% found in our sample contrasts with the 8.6% described in...., 368 a 21.2%, 369 an 11.0%, the 20%)
Response 5: Many thanks for that exhaustive reviewed; we have corrected those words at text.

Reviewer 2 Report
Dear authors,
this is a study on OA incidence and outcomes in Valencia region in Spain compared to general European population based on EUROCAT registry between 2009-2019.
I found it interesting for educative purposes but expected to find out more about etiology of the congenital anomaly from your manuscript.
In introduction section for educational goals you may want to create and insert a figure with comparison of Vogt's classification and ICD10-BPA e.g. with emphasis on group V by Vogt which seems not to fulfill nor Q39.0 nor Q39.1 criteria.
In line 98 give full term for LB first mentioned in the text.
You may want to compare OA perinatal outcomes with your general VR population to check the differences (table 2 and other birth data).
What is the explanation for 2 times lower prevalence of OA in Castellón province vs Alicante one and European data?
Since you found increase in prenatal diagnosis of OA over the 2 decades specify which ultrasound markers would you recommend in first, second and third trimester screening to further improve prenatal detection? Could you point any biochemical markers similar to AFP in spina bifida which might be helpful prenatally?
In discussion you may want to emphasize the value of your findings and additions to our knowledge about OA and TEF.
Thank you for your efforts to produce the paper.
Author Response
First of all, thanks for the comments about the paper. Regarding the review, we responded to your comments.
Point 1: In introduction section for educational goals you may want to create and insert a figure with comparison of Vogt's classification and ICD10-BPA e.g. with emphasis on group V by Vogt which seems not to fulfill nor Q39.0 nor Q39.1 criteria.
Response 1: Thanks for the idea to create a figure with comparison, we added the figure. In addition, in methods we added the specification about type V; it was not taking into account in this study because it doesn’t include OA diagnosis. And in results, we renumber the figures.
Point 2: In line 98 give full term for LB first mentioned in the text.
Response 2: Thanks for the review about the abbreviations. The term “LB” is mentioned in line 46 with the full term for LB.
Point 3: You may want to compare OA perinatal outcomes with your general VR population to check the differences (table 2 and other birth data).
Response 3: Thank you for your suggestion; we can’t compare OA perinatal outcomes with our general VR population, because this data is not available. But we can compare with the data about the congenital anomalies population-based registry of Valencian Region (RPAC-CV). We added these comparisons in the discussion. Consequently, we renumber the references.
Point 4: What is the explanation for 2 times lower prevalence of OA in Castellón province vs Alicante one and European data?
Response 4: Thank you for the review of prevalence data. The differences found between the provinces of the Valencian Region could be due to several factors, but to know this information would have to make studies where these variables of different exposures are included and the causality between them is studied.
Point 5: Since you found increase in prenatal diagnosis of OA over the 2 decades specify which ultrasound markers would you recommend in first, second and third trimester screening to further improve prenatal detection? Could you point any biochemical markers similar to AFP in spina bifida which might be helpful prenatally?
Response 5: Thank you for your suggestion, but this is not an object of our study. We only described the most frequent diagnostic technique during the prenatal stage.
Point 6: In discussion you may want to emphasize the value of your findings and additions to our knowledge about OA and TEF.
Response 6: Thank you for your suggestion, some modifications have been applied to the discussion.
